# Efficacy of Bioactive Glass Nanofibers Tested for Oral Mucosal Regeneration in Rabbits with Induced Diabetes

**DOI:** 10.3390/ma13112603

**Published:** 2020-06-07

**Authors:** Noha Elshazly, Abdelaziz Khalil, Manal Saad, Marco Patruno, Jui Chakraborty, Mona Marei

**Affiliations:** 1Tissue Engineering laboratories Faculty of Dentistry, Alexandria University, Alexandria 21526, Egypt; saad_manal@hotmail.com (M.S.); Mona.marei@dent.alex.edu.eg (M.M.); 2Oral and Maxillofacial Surgery, Faculty of Dentistry, Alexandria University, Alexandria 21526, Egypt; abdelaziz.khalil@outlook.com; 3Oral Biology, Faculty of Oral and Dental Medicine, Ahram Canadian University, Giza 12451, Egypt; 4Department of Comparative Biomedicine and Food Science, University of Padova, 35020 Legnaro (PD), Italy; 5Bioceramics and Coating Division, Central Glass and Ceramics Research Institutes, Kolkata 700032, India; jui@cgcri.res.in; 6Removable Prosthodontics Department, Faculty of Dentistry, Alexandria University, Alexandria 21526, Egypt

**Keywords:** oral mucosa, diabetes mellitus, diabetic induction, bioactive glass, nanofibers, wound healing, bioceramics, alloxan monohydrate, immunohistochemistry, oryctolagus cuniculus

## Abstract

The healing of oral lesions that are associated with diabetes mellitus is a matter of great concern. Bioactive glass is a highly recommended bioceramic scaffold for bone and soft tissue regeneration. In this study, we aimed to assess the efficacy of a novel formula of bioactive glass nanofibers in enhancing oral mucosal wound regeneration in diabetes mellitus. Bioactive glass nanofibres (BGnf) of composition (1–2) mol% of B_2_O_3_, (68–69) mol% of SiO_2_, and (29–30) mol% of CaO were synthesized via the low-temperature sol-gel technique followed by mixing with polymer solution, then electrospinning of the glass sol to produce nanofibers, which were then subjected to heat treatment. X-Ray Diffraction analysis of the prepared nanofibers confirmed its amorphous nature. Microstructure of BGnf simulated that of the fibrin clot with cross-linked nanofibers having a varying range of diameter (500–900 nm). The in-vitro degradation profile of BGnf confirmed its high dissolution rate, which proved the glass bioactivity. Following fibers preparation and characterization, 12 healthy New Zealand male rabbits were successfully subjected to type I diabetic induction using a single dose of intravenous injection of alloxan monohydrate. Two weeks after diabetes confirmation, the rabbits were randomly divided into two groups (control and experimental groups). Bilateral elliptical oral mucosal defects of 10 × 3.5 mm were created in the maxillary mucobuccal fold of both groups. The defects of the experimental group were grafted with BGnf, while the other group of defects considered as a control group. Clinical, histological, and immune-histochemical assessment of both groups of wounds were performed after one, two and three weeks’ time interval. The results of the clinical evaluation of BGnf treated defects showed complete wound closure with the absence of inflammation signs starting from one week postoperative. Control defects, on the other hand, showed an open wound with suppurative exudate. On histological and immunohistochemical level, the BGnf treated defects revealed increasing in cell activity and vascularization with the absence of inflammation signs starting from one week time interval, while the control defects showed signs of suppurative inflammation at one week time interval with diminished vascularization. The results advocated the suitability of BGnf as bioscaffold to be used in a wet environment as the oral cavity that is full of microorganisms and also for an immune-compromised condition as diabetes mellitus.

## 1. Introduction

Diabetes mellitus is a metabolic disorder that is characterized by a deficiency in insulin secretion or action resulting in hyperglycemia [1]. The World Health Organization reported in 2016, that it is a potentially morbid condition affecting 422 million adults aged above 18 years worldwide and it is expected to affect 592 million in 2035. It is classified into two types: insulin-dependent diabetes mellitus (IDDM) or type 1, and non-insulin dependent diabetes mellitus (NIDDM) or type 2 [2]. Egypt is one of the top 10 countries in the world in the number of diabetic patients, according to the International Diabetic Federation (IDF). In 2013, the IDF reported that about 7.5 million have diabetes. They also reported that 2.2 million Egyptians have undiagnosed pre-diabetes [3]. 

The increased glucose level in the bloodstream is the leading cause of many cardiovascular diseases, renal failure, infection, neural dysfunction, and limb amputation [2]. Furthermore, many complications in the oral cavity are also associated with diabetes, such as delayed wound healing as well as inflammatory soft tissue pathologies, including gingivitis and irreversible periodontitis. Decrease salivary secretion is often one of the main side effects of diabetes mellitus that leads to a high prevalence of tooth loss [4]. In a cross-sectional study that was conducted by Saini et al. [4] in order to correlate between diabetes mellitus and the percentage of oral mucosal lesions, it was reported that diabetic patients presented a higher prevalence of mucosal lesions along with oral precancerous lesions than healthy individuals. The researcher related that to immunological defects result from disorders of the endocrine system in the diabetic patients [4].

The oral cavity is a unique environment, in which wound healing occurs in the presence of warm oral fluids. It contains several microorganisms and specific molecular compositions that have a role in the healing process [5]. Normally, the mucosal wound healing process goes through similar phases as skin. It initially starts with the primary hemostasis phase, in which vasoconstriction of the injured blood vessels occurs. This activates the platelets that have an important role in the aggregation phase and coagulation phase. Besides, platelets produce chemotactic agents, cytokines, and growth factors, which attract different cell types to the site of injury and initiate the inflammatory reaction that serves to remove the debris and necrotic tissues. The inflammatory phase is followed by granulation tissue formation and re-epithelialization [6].

Re-epithelialization of the mucosal wound is the most important part of the wound healing process. It acts as an interface between the outer environment and the underlying connective tissues. Consequently, it also protects the underlying structure from the pathological microbes and other insults, such as that normally present in the oral cavity [7].

In diabetes, there is retardation in cell proliferation and migration [8]. This retardation is related to hyperglycemia [7,8]. Consequently, increased blood glucose level leads to obstructive vascular sclerosis and decreased blood flow [9]. This causes prolonged hypoxia, which leads to impaired wound healing [10]. It was also reported that hypoxia and hyperglycemia are the main factors in increasing the Reactive Oxygen Species (ROS), which causes Deoxyribo Nucleic Acid DNA and protein destruction [10]. The presence of these ROS distract the formation of blood vessels gap junctions [10]. Furthermore, they inhibit the chemotactic activity of neutrophils and monocytes at the time of injury, thus decreasing the number of infiltrated cells to the wounding site. This results in a diminishing of the innate immunity in the wound area [11,12].

Salivary gland hypofunction is one of the characteristic features of diabetes mellitus that affects the healing of mucosal tissue. It decreases the antimicrobial activity, buffering capacity, and self-cleaning action of the oral mucosa surface. It also disturbs the balance of oral flora. Furthermore, the diabetic patient is featured by increased glucose level in oral fluids. As a result, the amount of glutathione and melatonin are decreased. These two salivary components act as scavengers for the ROS, so their diminish induce more oxidative stress, causing protracted wound healing [13].

The need to cover the denuded oral lesions resulting from trauma, ulcers, and after surgical removal of oral tumours is common in oral and maxillofacial surgery. The use of autogenic soft tissue grafts is considered to be the gold standard for the treatment of non-healing mucosal defects [14]. This includes either the skin graft or the mucosal graft; however, both have their drawbacks [15]. Donor site morbidity and post-operative pain are among their most common complications [16]. The skin graft differs from the mucosal tissue in consistency, color, and occasional hair growth. Using vascularized skin graft to avoid the diminution of blood supply also has drawbacks, such as the associated skin appendages and keratinization, which makes it easily infected in this wet environment and heals by scar [14,16]. On the other hand, the use of mucosal graft from tongue, buccal, or labial mucosa is more suitable, but it is limited by the availability of the donor site and medical condition of the patient [15].

The field of tissue engineering has emerged spectacularly in the last 25 years in order to improve the quality of life on all of its aspects and to help in the regeneration as well as restoration of the integrity and function of damaged tissues. It is an interdisciplinary science that merges all of the scientific principles of biomaterials, cell biology, biochemistry, transplantation, and engineering innovations for the replacement of tissue or organ function. The new generation of tissue engineering aims to develop new three-dimensional (3D) natural and synthetic scaffolds, mimicking the extracellular matrix role, thus guiding cells recruitment, arrangement, and regenerative abilities. Biomimetic scaffolds are among the most promising and successful approaches that promote a high cellular response for the restoration of destructed tissues [17,18,19].

Bioceramic scaffolds with bioactive properties have expanded use in various hard and soft tissues engineering applications. Silicate bioceramics, with its superior bioactive, angiogenic, and antibacterial characteristics reveal high soft and hard tissue regeneration efficacy [20]. Besides, their antitumor properties, due to an easy fusion of released ions into the surrounding tissues [21].

Bioactive glass (BG), which was invented by larry Hench, 1969 [22], was mainly synthesized to reconstruct different types of bone defects in load and non-load bearing areas. Between different types of bioactive glass, silicate, and borate-based BG have significant importance in different soft and hard tissue applications [20]. Despite that, borate BG is shown to exceed the silicate BG in the level of bioactivity. As it reacts with the surrounding body fluids four to five times faster than silicate-based BG. It allows for complete conversion of the glass into hydroxyl carbonate apatite layer (HCA) and formation of intimate contact between the scaffold and the surrounding tissues [23]. Thus, borate containing BG is more favorable for use, particularly in soft tissue applications [24]. As such, the bone-bonding ability of BG is well known, although the important aspect of bioactive glass application for soft tissue repair is still far from being explored [25,26]. It is known that impaired skin wound healing is the most common cause of mortality among diabetic patients. In this regard, it has been observed that bioactive glass-based nanofibers mimic the microstructure of fibrin, might trap blood platelets, and allow the formation of a wound cover that could support the healing process [27,28]. Doped and undoped fibers markedly improve wound tissue healing through collagen fibers deposition, orientation, and maturity. In another study, BG introduced multifunctional gelatin/chitosan (G/C) nanofibrous membranes/mats have been used in chronic wound management [28]. A pioneering work conducted by Day [20] and his group used electrospun micro/nanofibers for chronic, non-healing wounds. The nanofibers provide cellular, mechanical support and aid in soft tissue regeneration and subsequent wound healing. Further, the large surface area helps to control the moisture level at the wound site. Additionally, the constituents of nanofibers activate the expression of genes that are related to wound healing, including cell-surface glycoprotein CD44 antigen, vascular endothelial growth factor (VEGF) precursor, etc., thereby stimulating neovascularization of the wound tissue [20]. 

In the present study, we succeeded to synthesize a novel bioactive glass composition of (1–2) mol% of B_2_O_3_, (68–69) mol% of SiO_2_, and (29–30) mol% of CaO via the low-temperature sol-gel route and fabricate (BGnf) via the electrospinning technique. Following fibers preparation, the in vitro physicochemical characterization (X-ray diffraction (XRD), field-emission scanning electron microscope (FESEM)) and degradation study were carried out in order to evaluate the scaffold bioactivity. Moreover, the assessment of the efficacy of BGnf in enhancing oral mucosal wound healing and angiogenesis was tested via histological and immunohistochemical analysis in rabbits with chemically induced type I diabetes mellitus. 

## 2. Materials and Methods 

### 2.1. Preparation of Borate Bioactive Glass Nanofibers

The sol-gel process was adopted for the synthesis of borate-based bioactive glass, as reported earlier by our group, followed by the fabrication of nanofibers by electrospinning process while using polyvinyl butyral (PVB) (BUTVAR^®^ B-72, Merk, Mumbai, India) [29] (Figure 1). Briefly, glass sol was prepared by adding 0.680 g Polyethylene oxide (PEO) Sigma Aldrich, Bangalore, India), 6 mL of tetraethyl orthosilicate (TEOS)( Sigma Aldrich, Bangalore, India), 3 mL of tributyl borate (TBB) (Sigma Aldrich, Bangalore, India), and 6 g of calcium nitrate tetrahydrate (CaNO_3_·4H_2_O)( Sigma Aldrich, Bangalore, India) sequentially into 1 N HCl solution. The glass sol was then mixed with a solution of PVB at a ratio of 3:2 to adjust its rheological properties for electrospinning. Following electrospinning, the glass nanofibers were subjected to heat treatment at 700 °C for 1 h at a heating rate of 2 °C/min. 

### 2.2. Characterization of Glass Fibres

The XRD patterns of bioactive glass nanofibers were obtained from X’Pert Pro MPD diffractometer (Panalytical, Almelo, The Netherland) while using CuKa (ƛ = 1.5418 Å) radiation at 40 Ma and 40 kV. Fibers were scanned at 2*Ө* range from 10°–80° with a step size of 0.03° and a count time of 2 s at each point. Nanofibers morphology and diameter were assessed using FESEM (Carl Zeiss SMT AG UPRA 35VP, Oberkochen, Germany) [30,31]. Before imaging, the glass fibers were sputter-coated with a 12 nm thickness of gold. An accelerating voltage of 10–20 kV and working distances of 5–8 mm were used. 

### 2.3. In Vitro Biodegradation

Static in vitro biodegradation of BGnf was done by soaking in a serum-free Dulbecco’s modified Eagles medium (DMEM) at a concentration of 0.0004 g/mL at 37 °C. Aliquots of the BG extract were collected at 24, 48, and 72 h time intervals. The BGs extract was analyzed using inductively coupled plasma atomic emission spectroscopy (ICP-AES) (Spectro Analytical, Kleve, Germany)in order to determine the amount of Si, Ca^2+^ ion release in the serum-free medium [32].

### 2.4. Sterilization of Glass Nanofibers

The nanofibers were sterilized for the first time by gamma-ray and then vacuum sealed. In re-sterilization UV light (wavelength of 200–280 nm) was used for 2 h in a cell culture safety cabinet (Telstar, Terrassa, Spain) that was equipped with UV [33,34].

### 2.5. Oral Mucosal Wound Defect Creation in the Diabetic Animal Model

#### 2.5.1. Animals

The Institutional Ethical Research Committee, Faculty of Dentistry, Alexandria University, Egypt reviewed and approved the study protocol (IRB NO: 00010556 IORG0008839, 24/9/2017). A total of 16 male New Zealand rabbits with an average age of 2.5–4 months and an average weight of 2–3 Kg were used. The rabbits were housed in windowed husbandry. The animals were individually kept under the same environmental conditions and continuous supervision. The rabbits were fed a restricted amount of commercial pelleted feed (133 g twice daily). A minimum temperature of 10 °C in the winter and maximum of 35 °C in the summer was maintained. A period of 12–13 h of daylight was provided. The animals were left for acclimatization for 10 days before the beginning of the experiment. The study followed the National Institutes of Health (NIH) guidelines for the care and use of laboratory animals [35]. The use of rabbits as an experimental animal model in the maxillofacial region has been reported being a good model for different studies on hard and soft tissues [36]. Rabbits are domestic animals that are easy to handle and observe. They are also cost-effective when compared to large animals, in addition to their short life cycle from gestation to puberty. This helps to manifest signs and chronicity of any disease over a short period. It is easy to provide daily care for these animals which have fewer health problems than other species [36]. Following the 3R principle (replacement, reduction, and refinement), sample size estimation was done with commonly estimated group standard deviations of 1 and with 95% confidence level and 80% power using independent *t*-test [37].

#### 2.5.2. Chemical Induction of Diabetes Mellitus in the Animal Model

Alloxan monohydrate (AMH) was used to develop diabetes mellitus in rabbits more than streptozotocin, which was mainly used to induce diabetes in rats as the alloxan is cost-effective and has a simple preparation technique [38]. It is a hydrophilic compound that is similar in its chemical structure to glucose. AMH causes selective destruction to beta cells by competition with glucose molecules and entering the beta cells on its Glucose Transporter 2(GLUT2) protein. As a result, AMH induces uncontrolled insulin secretion from beta cells as it inhibits the action of glucokinase which is considered beta cells glucose sensor. Also, it initiates ROS formation leading to beta cells damage. [39,40].

The induction of type I diabetes mellitus was done on 16 rabbits using AMH (Loba-Chemie, Mumbai, India). All of the procedures were performed under sterile conditions. Rabbits were allowed to eat and drink until the beginning of the induction procedure to avoid postoperative hypoglycemic shock [41]. The rabbits were anaesthetized using xylazine HCl (5 mg/kg) (Sigma-tech pharma industries, 6th of October City, Egypt), followed by ketamine HCl (50 mg/kg) (Rotexmedica, Trittua, Germany). The blood glucose level was measured before AMH injection using a standardized blood glucose meter (ACCU–CHEK Active, Roche Diagnostics GmbH, Mannheim, Germany) [41]. A single dose of 10% (*w*/*v*) AMH solution was achieved by dissolving 150 mg/kg body weights in sterile normal saline. The solution was injected via the marginal ear vein using 26 gauge cannula (Lars medicare, Haryana, India) over a period of 30–60 s [42] (Figure 2a,b). After complete recovery, the animals were transferred back to their cages and food and water were offered immediately. The blood glucose level was monitored every one hour for the first 7 h post-injection. Rabbits were subcutaneously injected with 20 mL of 5% glucose solution three to five times during the first two days to avoid post-operative hypoglycemic shock [42]. Contentious monitoring of blood glucose level was done every day for the first three days. After that, blood glucose was measured once weekly until the end of the experiment. The rabbits were confirmed as diabetic if the blood glucose level remained >200 mg/dL for 15 days after induction [41,42]. Following animal euthanization, sections of pancreatic tissues were histologically examined [43].

#### 2.5.3. Surgical Procedure

After two weeks of diabetes induction, 12 male New Zealand Rabbits were prepared for surgical operation under general anaesthesia, as mentioned before. The blood glucose level was measured immediately before surgery (Figure 2c). For each rabbit bilateral elliptical mucosal defects (10 × 3.5 mm) were created in the maxillary mucobuccal fold anterior to rabbit molars [44] (Figure 2e). Six randomly selected rabbits were considered as a control group, where the defects (n = 12) were left to heal without any intervention. The remaining six rabbits were considered as the experimental group, where defects (n = 12) were grafted with (0.087–0.09 g) bioactive glass nanofibers (Figure 2f). Each rabbit received an intramuscular injection of antibiotic (cefotaxime 1 g in a dose of 150 mg/kg/day) and analgesic (ketorolac 60 mg/day) for three days postoperatively. The soft diet was offered for three days post-operatively. Daily monitoring of animals was done until the end of the study [45].

### 2.6. Histological and Immunohistochemical Evaluation

#### 2.6.1. Sample Collection for Histological Analysis

The collection of mucosal biopsies was done after the rabbit’s euthanization using an overdose of xylazine followed by cervical dislocation. Full-thickness mucosal biopsies were harvested up to the bone surface from the operated region of the maxillary mucobuccal fold, at one, two, and three weeks’ time intervals. The biopsy included safety margins, extending from the distal side of central incisor to the mesial side of the first premolar for histological examination. The samples were fixed in formalin 10% and then dehydrated in graded series of different concentrations of ethanol (70%, 80%, 90% and 100%). Dehydrated samples were cleared while using xylene, embedded in paraffin wax, and were then cut in 7 µ thin sections. The tissue sections were stained with haematoxylin and eosin (H&E) (Biotec, Chelopech, Bulgaria) to be examined and photographed by light microscopy (Olympus, Tokyo, Japan) [46].

#### 2.6.2. Immunohistochemistry

For immunohistochemical staining, 5 μ thin tissue sections of both groups were deparaffinized and rehydrated. Tissue sections were then incubated in Hydrogen Peroxide Block for 10 min. at room temperature in order to reduce nonspecific background staining. Sections were then washed with distilled water and PBS. Tissue sections were immunostained for the VEGF (Thermo Fisher Scientific, Waltham, MA, USA). Immunostaining was done using mouse anti-rabbit monoclonal antibodies, while control sections were prepared following the same staining steps without including the primary antibodies. The antibodies were diluted using 3% BSA-PBS (Thermo Fisher Scientific, Waltham, MA, USA) at a dilution of 1:100 and sections were incubated for 1 h at room temperature. Tissue sections were washed extensively in the buffer. Horse Radish Peroxidase (HRP) (Thermo Fisher Scientific, Waltham, MA, USA) polymer-conjugated secondary antibody- was added for 15 min. at room temperature, followed by colorimetric detection while using DAB kit (Thermo Fisher Scientific, Waltham, MA, USA). Counterstaining was done using haematoxylin. All of the immunostaining steps were performed using bond III fully automated immune-histochemical stainer (Leica Biosystem, Wetzlar, Germany) [47,48]. An expert histologist blindly assessed analyses of sections by selecting four random non-overlapping fields (20× magnification) per tissue section. Each was analyzed for calculation of percentage area with an expression of VEGF [48]. All of the micrographs and data were obtained by using Full HD microscopic camera (Lecia Microsystems GmbH, Wetzlar, Germany) operated by application module for tissue sections analysis (Leica Microsystems GmbH, Wetzlar, Germany).

### 2.7. Statistical Analysis

Statistical analysis was performed using SPSS16.0. Statistical differences were evaluated using the paired and unpaired *t*-test. Significance was considered when *p* ≤ 0.05. Means and standard deviations were calculated for all variables in all groups.

## 3. Results

### 3.1. Characterization of Glass Fibers

#### 3.1.1. X-ray Diffraction (XRD)

Analysis of the BGnf pattern showed a broad peak around 20° to 23° corresponding to the Si-O-Si network indicating the amorphous structure of the glass nanofibers [29,49] (Figure 3).

#### 3.1.2. Morphological Analysis of the Glass Fibers

Based on FESEM images, borate bioactive glass nanofibers showed an ultra-structure of crosslinked nanofibers with a different range of diameter (500–900 nm) [29] (Figure 4b). Nanofibers architecture appeared to simulate that of the natural fibrin clot formed during the primary stage of the wound healing cascade (Figure 4a).

#### 3.1.3. In-Vitro Biodegradation

The degradation profile of BGnf showed a gradual increase in the rate of silicon and calcium ions release in correlation with the time. The dissolution of ions reaches a high level after 72 h, indicating the rapid reaction of glass nanofibers with the surrounding tissue fluids (Table 1).

### 3.2. Diabetic Induction in the Rabbit Animal Model

Following alloxan injection, rabbit’s blood glucose level was found to increase from the basic normal range of 127–140 mg/dL before induction to as high as 300–350 mg/dL, 72 h following induction. The increased blood glucose level was sustained throughout the study period (Figure 5). Out of the 16 rabbits that were subjected to alloxan injection, 12 rabbits completed the whole study. The other three rabbits died due to severe hypoglycemia after alloxan injection. Moreover, the failure of diabetic induction in one rabbit was also noticed (Table 2). On the histological level, an evaluation of the pancreatic tissue of diabetic rabbits showed atrophic changes in the size of islets of Langerhans (Figure 6b). The nuclear degeneration of Langerhans cells was also observed after two weeks of diabetic induction. On the contrary, normal pancreatic tissue revealed active deeply stained nucleus of Langerhans cells [43] (Figure 6a).

### 3.3. Effect of BGnf on the Full-Thickness Mucosal Wound

#### 3.3.1. Macroscopic Observation 

Following surgery, no clinical signs of hematoma or severe inflammation were detected in both of the study groups. After one-week time interval, BGnf treated wound showed complete wound closure with the restoration of tissue texture and absence of clinical signs of inflammation (Figure 7a). Greyish shadow at the center of BGnf treated wound was also observed. On the other hand, control wound showed incomplete wound closure as well as the presence of purulent exudate in the center of wound defect (Figure 7b). Both experimental and control animal groups of two and three weeks’ survival periods revealed macroscopic wound closure. Fainting of the greyish shadow in the BGnf treated wound was noticed at three weeks’ time interval (Figure 7c). The mucosal tissue of the control wound revealed tissue projection in the wounding area (Figure 7d).

#### 3.3.2. Histological Assessment

The histological results of the BGnf treated and untreated wound defects of diabetic animals are represented in (Figure 8). BGnf grafted mucosal wound at one week time interval is illustrated in (Figure 8a,g), showing the regeneration of epithelial cell layers that have not yet reached their full thickness(Figure 8g). Dense collagen fibers with regular vascular distribution were also observed running parallel to the overlying epithelium (Figure 8g). In contrast, the control wound defect showed absence of epithelial lining at the center of the defect (Figure 8b,j). Loose connective fibers were seen extended in a horizontal direction with perivascular cell infiltration of inflammatory cells indicating the suppurative inflammation (Figure 8m). This corresponded to the clinical picture observed in the defect of the same group of animals (Figure 7b).

At two weeks’ time interval, BGnf treated wounds showed complete closure of epithelial lining in the center of the wound defect along with prominent basal cell layer indicating high cellular activity (Figure 8c,h and Figure 9a); however, it was thinner than the surrounding epithelial tissue (Figure 9c). The connective tissue layer of lamina propria revealed the absence of inflammatory reaction and regular vascular distribution (Figure 9b). While control wound at the same period was showing creeping of the epithelial cells toward the center of the wound defect with the absence of inflammatory cell infiltration and diminished vascular supply (Figure 8d,k). At three weeks’ time interval, BGnf treated wound revealed normal epithelialization with mature connective tissue matrix (Figure 8e,i). On the other hand, the control wound was still showing incomplete wound closure with cell migration from the surrounding epithelium toward the center of the wounding area (Figure 8f, l). Maturation of the connective tissue layer of lamina propria was observed (Figure 8l). Obvious vascularization and cell infiltration from the deep submucosal layer were seen in BGnf treated wounds during previously mentioned time intervals, in addition to traces of BGnf (Figure 9d). These traces were encapsulated and surrounded by different types of cell infiltration as fibroblast-like cells and phagocytic cells (Figure 9c,d). Multinucleated giant cells with engulfed material traces were found surrounded by a high number of newly formed blood vessels (Figure 9c–e).

#### 3.3.3. Immunohistochemical Analysis

VEGF Expression

After one and three weeks’ time interval, the immunohistochemical expression of VEGF was assessed in the BGnf treated group and control one. At one week time interval, significant (*p* < 0.005) higher intensity of VEGF expression was found in lamina propria layer in the BGnf treated mucosal wounds than that of control wounds (Figure 10) (Table 3). At three week time interval, an increase of VEGF expression was observed in both BGnf and control wounds, However, BGnf treated wounds remained significantly expressing a higher intensity of VEGF when compared to the control one (Table 3) (Figure 10e).

## 4. Discussion

Protracted wound healing is considered to be a serious complication of diabetes mellitus as; it leads to the destruction of the outer tissue barrier, which protects the inner tissues from infection [6,7]. This is highly related to the decrease in vascular supply that results from increasing accumulation of advanced glycosylation end products (AGEs) in vascular walls [9]. Moreover, it is reported that diabetes causes a significant decrease in growth factors, such as VEGF and Platelet-Derived Growth Factor (PDGF). Besides, it hinders the migration and proliferation of endothelial cells, fibroblast cells, as well as epithelial cells at the region of tissue injury [5,50].

Skin and oral mucosa proceed through similar stages of wound healing process, starting from hemostatic phase reaching the remodelling phase [51]. They differ in that oral keratinocytes and fibroblasts reveal fetal tissue like features, so they proliferate lightly higher than dermal cells leading to faster wound healing with less scar formation [52,53]. Additionally, the lower expression of VEGF in oral mucosal wounds was observed while comparing to that of the skin wounds [54]. The presence of saliva in the oral cavity is another difference, as it provides the humid environment that is essential for accelerating cell proliferation and migration; it also allows oxygen and nutrient supplies to deliver faster to tissue cells [55]. Besides, saliva contains different types of proteins and peptides that have a direct and indirect impact on accelerating mucosal wound healing [55]. On the other side, saliva provides the oral cavity with the optimal conditions that are suitable for complex microbiota growth, being composed of more than 1000 various species of bacteria, fungi, and viruses [56].

Bioactive glass is composed of different metal oxides formulas that dissolute in tissue fluids, leading to the formation of HCA, allowing for intimate contact between the glass scaffold and the surrounding hard or soft body tissues. Moreover, it allows the adsorption of growth factors from surrounding tissue as well as enhancing cell attachment, proliferation, and migration [22,24]. Multiple types of bioglass have been recently introduced, depending on the variability in its metal oxides compositions. Silicate and borate BGs are among the famous types used in tissue engineering applications [20,24]. In the present study, novel 3D bioactive glass electrospinned nanofibers composed of (1–2) mol% of B_2_O_3_, (68–69) mol% of SiO_2_, and (29–30) mol% of CaO were used to regenerate mucosal defect in rabbits with induced diabetes.

It was reported by different studies that electrospinning fibres can provide high surface area with a 3D architecture that was morphologically similar to that of ECM of body tissues, which would significantly improve cell attachment, proliferation, and migration [57,58]. Another study also reported that the electrospinning nanofibres have nano range diameter, an elevated level of porosity, and mechanical properties, such as tensile and shear strength [59]. Kim et al. [60] were the first to synthesize bioactive glass electrospinning nanofibers that showed a significant increase in surface area, bioactivity, and in-vitro osteogenic properties. Moreover, Deliormanlı, 2015 [30] used the electrospinning technique to prepare 45S5 bioactive nanofibers having a nano diameter that ranged from 280 to 340 nm, with a high porosity level along with high ionic degradation rate. Our data showed nearly similar results where the produced electrospinning nanofibres had a 3D cross-linked nanofibrous architecture that was similar to that of a fibrin clot with fibres diameter ranged from 500 to 900 nm. Ma et al. [28] also reported similar results during preparing BG/Gelatine nanofibers with different concentration. He reported that, although the formed fibres had different morphology, they were tens-to-hundreds of nanometers in diameter. This diameter was similar to that of natural collagen matrices [28].

XRD analysis of the formed fibres in the present study confirmed its amorphous structure, which was comparable to a study completed by Zhao et al. [49], which showed an extended band concentrated between 20–30° identical to the amorphous structure. The importance of amorphous nanofibrous structure is that it provides a large surface area and ECM-like organization that is suitable for platelet attachment, cell adhesion, as well as cell proliferation and migration. It also allows for nutrient and oxygen exchange, which all play a role in accelerating the wound healing process [49,59].

In the present study, the degradation profile of BGnf showed a higher dissolution rate of Ca^2+^ and Si ions, starting from 24 h and reached its highest level after 72 h. The study that was conducted by Saha et al. [29] also showed that the BGnf ionic dissolution rate reached its highest value after 72 h; following that, it reached to a static state after 168 h that might be related to the ionic saturation of the surrounding media. Ionic dissolution contributes to the level of BG bioactivity. Silicate glasses require months to years in order to fully turn to an HCA layer [61]. On the contrary, borate containing bioactive glasses form the HCA layer directly when contacting body fluids [20]. Jung et al. [23] also confirmed this, as it was found that borate glasses react four to five times faster than the silicate equivalent in body fluids. This was highly related to raising solubility of borate in tissue fluids. The result of our study was comparable with these previous studies. It was also comparable with the result of another study, in which 13-93B3 bioactive glass microfibers showed the total conversion of the immersed BG fibres into HCA within seven days of immersion in simulated body fluid [49]. 

Bioactive glass nanofibrous structure mimics fibrin clot and allows for moisture control as well as platelet aggregation at the site of wound defect [62]. In addition, the ionic products of BG also play an important role in wound healing processes. It has been reported that calcium ions activate the secretion of thrombin as well as fibrin at the area of wound defects in the early stages of clot formation [63]. They also control different genes that are responsible for epithelial migration [62,64]. Meanwhile, silicate ions contribute to the angiogenic activity of endothelial cells, as they enhance endothelial cell proliferation and activate secretions of multiple growth factors, like VEGF and bFGF from tissue fibroblast [65,66]. Moreover, it was reported that boron ions have a high tendency to stimulate the translation of encoding mRNA growth factors that are responsible for angiogenesis and wound healing, such as VEGF and TGF-β [67]. The antibacterial activity of the BGnf was suggested to be related to the rapid elevation of pH values from 7 to nearly 9 and osmotic pressure in the tissue fluids as a result of ionic dissolution, which results in the inhibition of bacterial growth [28,68]. 

The present study confirmed the previously mentioned findings as the BGnf mucosal treated wounds in diabetic animal models showed epithelial cell migration to the center of the wound defect as well as deposition of collagen fibres after one week time interval. Histological analysis of the BGnf grafted wounds revealed its superiority over the control wound on the rate of wound healing process, starting from granulation tissue formation, collagen fibres arrangement, as well as the migration and proliferation of the epithelial cells to the defect site. These data were highly correlated with previous results that were reported by Saha and his group that revealed the tendency of BGnf extraction media to accelerate in vitro cell migration and proliferation rate at 24 h with a percentage that reached 82% when compared to the 47% migration rate of the control group [29]. Another study was conducted by Zhou et al. [68] to compare fish col/BG nanofibers and fish col. nanofibers, as wound healing scaffolds showed a higher proliferation rate of HACAT cell line seeded on fish col/BG nanofibers. In addition to that, the study showed a significant improvement in the mechanical, antibacterial, angiogenic, and wound healing properties of col/BG nanofibers when compared with the fish col. nanofibers alone. The borate element that has been added to BG scaffold in the present scaffold has already demonstrated its wound healing efficacy in several studies [49,62,69].

The expression of VEGF showed a significant increase in the BGnf treated wound at one and three week time intervals when compared to that of the control wound that indicated the increased angiogenic activity in the experimental wounds. VEGF was reported to increase fibroblast activity and collagen fibres deposition [66]. The angiogenic efficacy of BGnf was reported in different in vitro and in vivo studies [59,68,70]. On the other side, it was reported that the Cu dopped BGnf showed higher angiogenic activity when compared with the non-doped one [49].

Glass nanofibers were reported to enhance collagen I secretion from human dermal fibroblast cells (HDFs). Besides, on the histological level, it was observed that BGnf stimulated rapid deposition of collagen fibres in dermal tissue during the early stages of skin wound healing [68]. Xu et al. [59] also mentioned these results, in which the patterned BGnf treated wounds showed a higher deposition of organized collagen bundles in the dermal layer than the control wounds with increased Collagen I and III secretions after at one week time interval. Moreover, Zhao et al. [49] showed similar results in skin wounds that were treated with Cu doped BGnf and non-doped one, in which the laydown of mature and organized collagen bundles were observed. The histological observations of the present study were equivalent to the previous studies, as the BGnf treated mucosal wounds have shown denser collagen bundles in the lamina propria layer when compared with the loose collagen fibers deposition in control wounds, starting from one week time interval.

In this study, the absence of clinical and histological signs of infection was noticed in the BGnf treated wound, while the control wounds expressed purulent exudates and the accumulation of inflammatory cells in the lamina propria layer, indicating suppurative infection. Diminished immunity is highly related to diabetes mellitus, as was mentioned before due to decreased blood supply and increase insulin resistance. The present study confirmed the antibacterial activity of BGnf that was mentioned in different studies [28,29,68,69].

From our clinical point of view, BGnf is considered an excellent grating material to be regularly used as a treatment for gum tissue defects. It is a cost-effective, easily applied scaffold. Besides, it can be adapted to the wound area and compacted inside. In addition to that, the scaffold can be applied multiple times without any difficulties. Other studies also mentioned similar findings that were related to the ease of material application and its cost-effectiveness [62,69]. 

One disadvantage that was observed regarding BGnf was the greyish discoloration of the mucosal tissue at the site of the wound. We suggested that it is highly related to the BGnf remnants that accumulated in the deep submucosal layer circumscribed by the multinucleated giant cells. It might also be related to the thin thickness of oral epithelium with the absence of keratinized layer. At three weeks’ time interval, the greyish discoloration started to disappear to a degree that it would not be noticeable. This has been confirmed in a previous study upon using a commercially available BGnf to treat unhealed chronic skin injuries in canine patients [69], where the observed discoloration was of no effective clinical significance. 

To our knowledge, this is the first study to use BGnf to treat oral mucosal defects in a diabetic animal model. The used BGnf was also composed of novel concentration of the metal oxides. The method that was used to induce diabetes mellitus in the rabbit animal model by alloxan monohydrate was accomplished after reviewing the literature and a preliminary study, as a minimal number of studies used rabbits as a model for diabetic research. In this study, we combined both techniques that were described by Wang et al. [41] and Oh et al. [42], in which we allowed animals to eat and drink before injection to avoid the postoperative hypoglycemic shock that proved to reduce the rate of animal death. Moreover, using a 10% solution of 150 mg/kg alloxan monohydrate enabled hyperglycemia after a single dose only, which was sustained until the end of the study. In agreement with a previous study that reported microscopic changes of pancreatic tissues after AMH injection [43], our histological observations confirmed the degenerative effect of the AMH on islets of Langerhans with minimal effect on the exocrine glands.

One limitation has been encountered in the present study, which was the difficulty to measure the decrease in wound size, as this would have required anaesthetizing the animal during wound examination that could have led to animal death by ketamine toxicity. Accordingly, for further investigation of the role and mechanism of action of BGnf in tissue regeneration, our group plans to study the wound healing of full-thickness skin defects on normal and diabetic animals while using this novel biomaterial.

## 5. Conclusions

The present study reports the synthesis of a novel formula of BGnf and illustrates its efficacy in accelerating gum tissue wound healing. In vitro characterization of BGnf indicates its amorphous architecture with an ultra-structure that simulates fibrin clot. Further, the ionic dissolution products of the BGnf shows higher degradation rate that advocates raised bioactivity of the used formula. BGnf accelerates oral mucosal wound healing in rabbits with induced diabetes. BGnf enhances collagen fibers lay down and epithelial cell migration to the defect area at a short time. Moreover, BGnf prevent bacterial invasion to the wounding area providing a sterile wound bed. VEGF expression is significantly increased in the BGnf grafted wounds when compared to the control wounds at one and three weeks’ time interval. Thus, the present study has highlighted BGnf as an effective grafting biomaterial for the treatment of gum defects.

## Figures and Tables

**Figure 1 materials-13-02603-f001:**
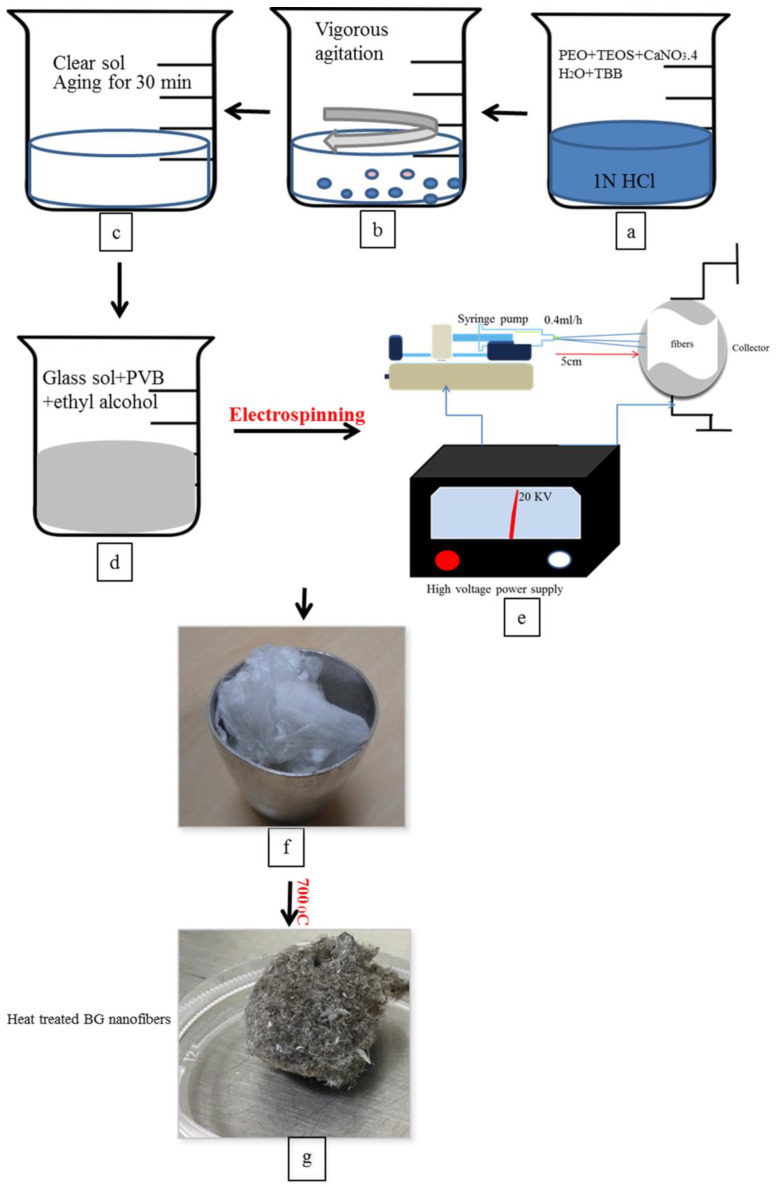
Diagram of borate bioactive glass nanofibres preparation. Where (**a**–**c**) represent the preparation of bioactive glass sol at room temperature. (**d**) Formation of polymer/glass mixture. (**e**) Diagram of electrospinning machine used to produce polymer/glass nanofibers. (**f**) Unheat-treated polymer/glass nanofibres. (**g**) Glass nanofibers after heat-treatment at 700 °C and retrieval of polymer remnants.

**Figure 2 materials-13-02603-f002:**
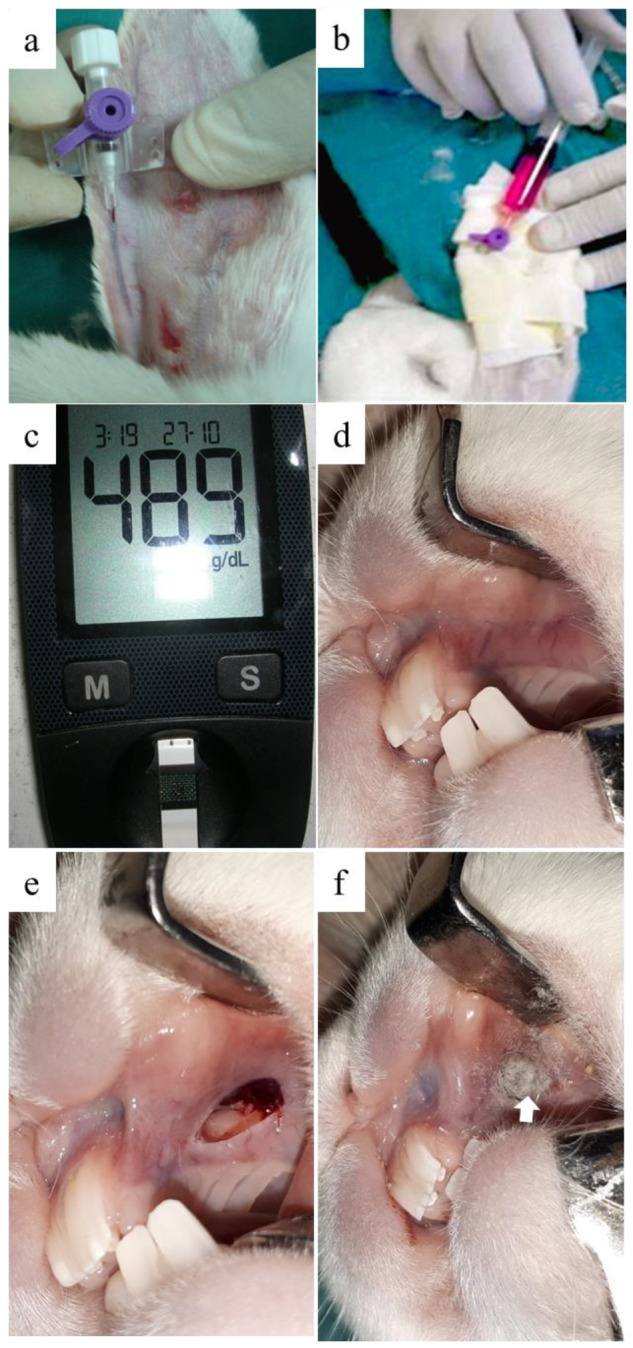
Clinical pictures of chemical induction of diabetes mellitus and the creation of full-thickness alveolar mucosal defect. (**a**) Insertion of 26 gauge cannula in rabbit’s marginal ear vein was followed by (**b**) injection of a single dose of 10% alloxan monohydrate solution. (**c**) Measurement of blood glucose level after two weeks ensuring diabetic induction. (**d**) Healthy mucosa. (**e**) 10 × 3.5 mm full-thickness elliptical defect in the maxillary alveolar mucosa was surgically created. (**f**) Grafting of the experimental group of mucosal defects using BGnf (arrow).

**Figure 3 materials-13-02603-f003:**
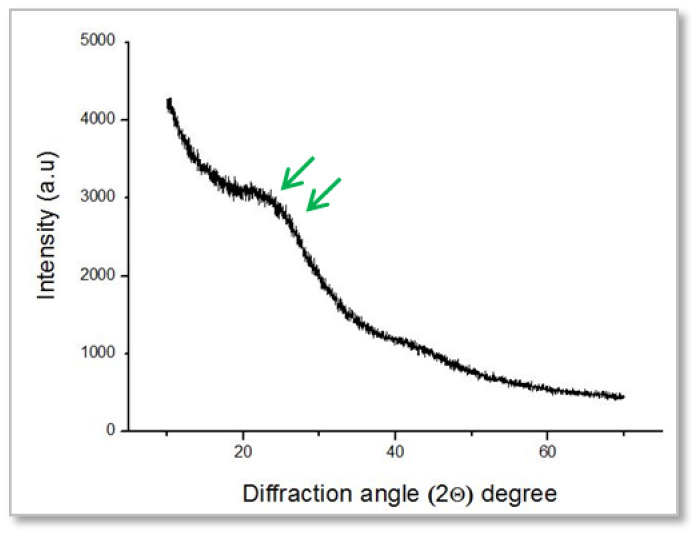
Powder X-ray diffraction (PXRD) pattern of BGnf, showing a broad peak around 20° to 23° indicating the amorphous structure of the glass nanofibers.

**Figure 4 materials-13-02603-f004:**
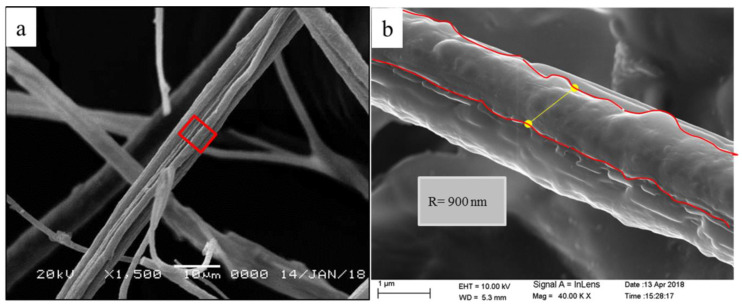
Characteristic features of borate bioactive glass nanofiber (BGnf), where; (**a**) scanning electron microscopy photograph at low magnification (10 μ) showing the microscopic structure of BGnf that highly simulating the natural fibrin clot. (**b**) Scanning electron microscopy photograph at a higher magnification (1 μ) illustrating the nano-range of the formed fibers.

**Figure 5 materials-13-02603-f005:**
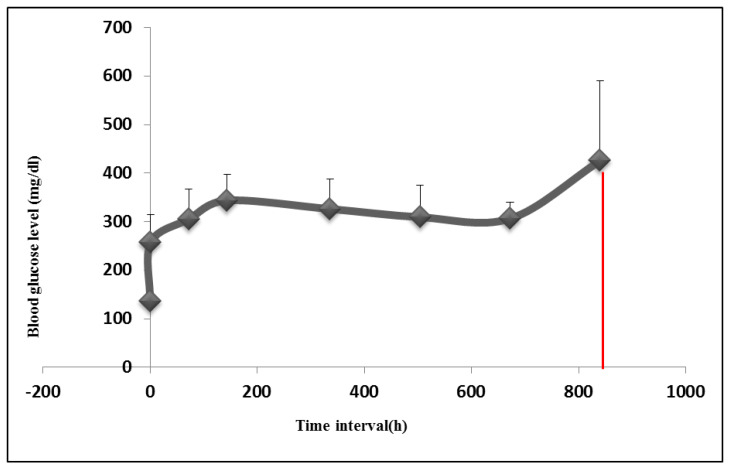
Graphical representation of the mean value of blood glucose level of both groups of rabbits throughout study periods. Starting from 0 h before diabetic induction until the end of the study (five weeks after induction (850 h) blood glucose level showed a steady increase in the blood glucose level throughout the study period.

**Figure 6 materials-13-02603-f006:**
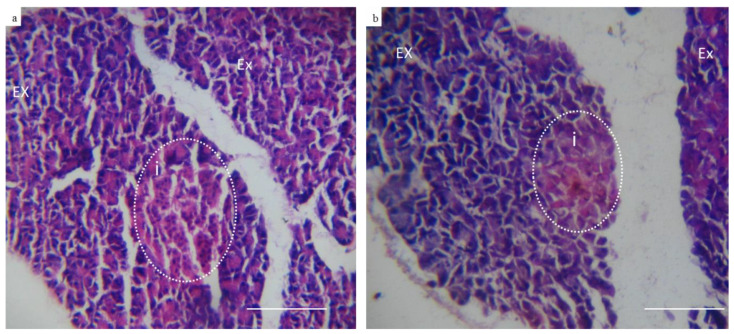
Representative photomicrographs of control and diabetic rabbits pancreas (Hematoxylin & Eosin) (10×). (**a**) Showing control pancreatic tissue section of well-organized exocrine glands (EX) and islets of Langerhans (i) which are formed of cells with deeply stained nucleus indicating its high activity. (**b**) Representing pancreatic tissue section retrieved from rabbit with type I induced diabetes showing unorganized exocrine glands (EX) and islets of Langerhans with atrophic changes and degenerated cells (absence of nucleus) indicating loss of excretory function.

**Figure 7 materials-13-02603-f007:**
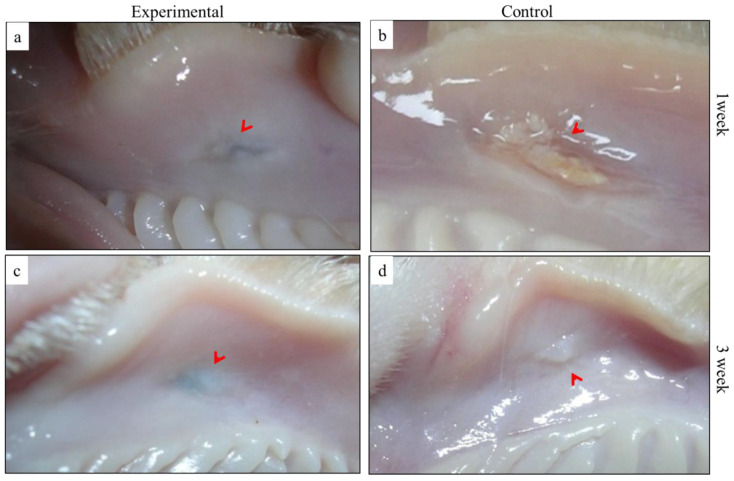
Clinical appearance of the operated defect at one and three weeks’ time intervals showing the obvious difference between (**a**) BGnf treated wound; with complete wound closure and restoration of the mucosal texture (arrowhead) and (**b**) the unclosed purulent defect of the control wound at one week time interval. At three week time interval (**c**) the BGnf treated wound showed better tissue remodeling with temporary discoloration that related to the presence of remnant of BGnf (arrowhead) versus (**d**) projected region of the healing mucosa of the control wound at the same time interval (arrowhead).

**Figure 8 materials-13-02603-f008:**
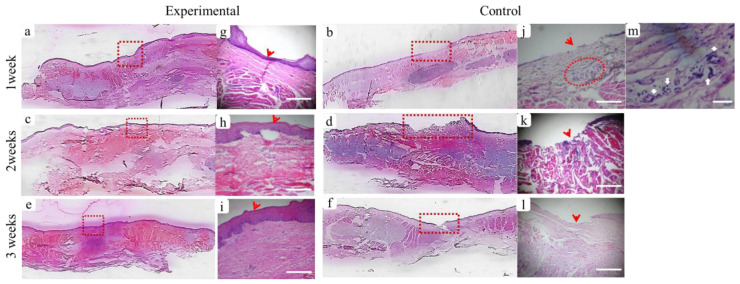
Panoramic (Hematoxylin-Eosin) histological sections showing full-thickness mucosal defects to compare between wounds treated with borate-based bioactive glass nanofibers (BGnf) (experimental defect; **a**,**c**,**e**) and untreated mucosal defect (control defect; **b**,**d**,**f**) at different time intervals. (**a**) Panoramic view of BGnf treated defect at one week time interval showing wound closure by re-epithelialization (square area). (**g**) Higher magnification of the BGnf treated wound at the same time interval representing regeneration of basal epithelial cell layer (arrowhead) (scale bar: 500 µm), while (**b**) control wound was still showing missing epithelial layer at one week postoperatively (rectangular area). (**j**) Higher magnification of the control wound at the same time interval (scale bar: 500 µm) showing incomplete epithelial formation (arrowhead) and the presence of perivascular cell infiltration in the lamina propria (circular area). (**m**) Higher magnification of the perivascular area (circular area in j) in which infiltration of different inflammatory cells was observed (arrows) (scale bar: 100 µm). (**c**,**e**) Representing panoramic views of BGnf treated defects at two and three weeks postoperatively showing progression in wound healing that was highlighted in (**h**,**i**) respectively in the form of increasing in epithelial thickness, as well as collagen fibers maturation in the underlying lamina propria (square areas). (**d**,**f**) Panoramic views of the control wounds at two and three weeks’ time intervals were still showing incomplete epithelialization at the center of wound defects (rectangular areas) which were highlighted in (**k**,**l**) respectively (scale bar; 500 µm).

**Figure 9 materials-13-02603-f009:**
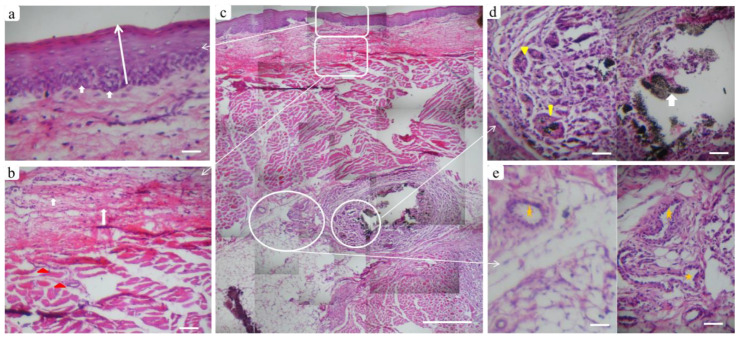
Tissue reaction in the BGnf treated defect two weeks postoperatively. Where (**c**) panoramic view of multiple serial histological sections of full-thickness mucosa at the defect site (scale bar: 500 µm) to show tissue response from the outermost epithelial surface reaching to the submucosal layer. (**a**) Higher magnification of the epithelial layer, where the normal stratified squamous appearance of epithelium could be seen with prominent basal cell layer (arrows). (**b**) Lamina propria layer showing newly formed collagen fibers (arrows) extending to the deeper submucosa with a regular distribution of blood vessels (triangles). (**d**) Well circumscribed remnants of BGnf (arrow) that were observed in the deep submucosal layer surrounded by inflammatory cells and engulfed by multinucleated giant cells (triangles). (**e**) Prominent blood vessels were distributed in the same region (stars) indicating BGnf high angiogenic activity (scale bar: 100 µm).

**Figure 10 materials-13-02603-f010:**
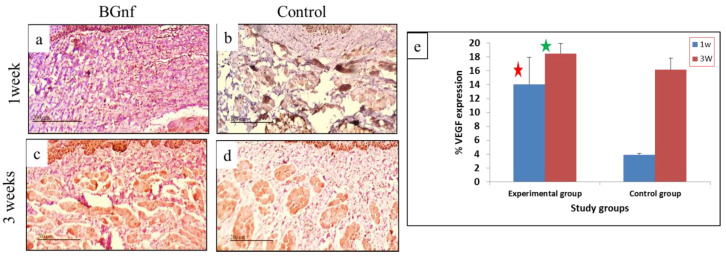
Microscopic photographs and Graphical representation of the immunohistochemical expression of VEGF in BGnf treated and control mucosal wounds at one and three weeks’ time interval using DAPI. (**a**,**c**) Showing higher color intensity at the lamina propria of BGNF treated wound at (**a**) one week and (**c**) three weeks indicating high VEGF expression. (**b**,**d**) Showing VEGF expression in the control wounds at one week (**b**) and three weeks’ (**d**) time interval with observed lower intensity of brown coloration at the lamina propria layer indicating less VEGF expression compared to the experimental wound. (**e**) Graphical representation of the mean percentage of VEGF expression of the BGnf treated wounds and control wounds at one and three weeks’ time intervals, revealing significant higher intensity (*p* < 0.005) in VEGF expression (red and green star) of the BGnf treated wound at both time intervals when compared to that of the control wounds.

**Table 1 materials-13-02603-t001:** The degradation rate of BGnf at different time points.

Time (h)	BGnf Ionic Extract (ppm)	
	Si	Ca
24 h	41.25 ± 0.94	25.15 ± 1.28
48 h	60.36 ± 1.05	27.36 ± 1.45
72 h	67.35 ± 1.58	30.25 ± 1.07

**Table 2 materials-13-02603-t002:** Demographics of diabetic induction.

Total N.	Diabetic Induction			Study Groups	N	%
	Condition	N	%			
16 New Zealand white rabbits	Induced	12	75.0	Experimental	6	37.5
	Failed	1	6.25	Control	6	37.5
	Died	3	18.75	Total	12	75

**Table 3 materials-13-02603-t003:** Comparison of Percentage of vascular endothelial growth factor (VEGF) expression in oral mucosa tissue samples between BGnf treated wounds and control wounds.

Number	Percentage of VEGF Expression of BGnf Treated and Control Groups at 1 and 3 Weeks			
	One week		Three weeks	
	BGnf	Control	BGnf	Control
Nu of analyzed Fields	12	12	12	12
Mean ± SD	14.08 ± 3.88	3.92 ± 0.221	18.48 ± 1.458	16.81 ± 1.65
Sig. (2 tailed) P	0.0001 *		0.0064 *	

* Statistically significant *p* ≤ 0.05.

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
