# Peer review of "Efficacy of Bioactive Glass Nanofibers Tested for Oral Mucosal Regeneration in Rabbits with Induced Diabetes"

_materials, 2020, doi:10.3390/ma13112603_

Round 1

Reviewer 1 Report

The work deals with the preparation of a bioactive glass composition consists of B2O3, SiO2 and CaO via sol-gel and electrospinning methods to enhance oral mucosal wound healing. In the study, male rabbits were subjected to type I diabetic induction using alloxan monohydrate and prepared bioactive glass was placed in the wound to explore the healing performance in vivo. The paper is well structured and procedures are well described, supported by results.

However, some minor corrections are needed before publishing, as follows:

C1) Line 33: A space needed between and3. There are so many tied letters in the manuscript, especially before/after the brackets. It would be better to check it all.

C2) Lines 51, 81, 114, 216 : The (IDF), (ROS), (Larry Hench, 1969), (3R) does not require brackets since all were defined previously.  

C3) Line 88 : The D here supposed to be written in lower case “Furthermore, Diabetic patient…”

C4)  In Figure 1d, PVA is written, however, polyvinyl butyral  was reported in section 2.1 as used material. It needs to be fixed for both figure and text, either the used material was polyvinyl butyral (PVB) or polyvinyl acetate (PVA).   

C5) Lines 191 and 352 : The sample name of “ 70SBG” supposed to be the same sample with the sample of 70S 30 Ca BG. One of these needs to be fixed.

C6) Section 3.1.1 : The obtained peak corresponding to the Si-O-Si network, indicates porous structure (line 321) or amorphous structure (line 331) ? This need to be fixed and a reference needed as an indicator of the presence of Si-O-Si is indicating either porosity or amorphous structure. Also, “t0” supposed to be “to” (Line 330).

C7) In Figure 4b, the scale shows 1um and if it is correct, the R is bigger than 560 nm, compared to the scale.  This needs to be checked to correct either scale or the R value of the fiber, which looks slightly bigger than 2 um.

C8) Line 628 : The SBF, supposed to stand for the simulated body fluid. If so, it needs to be indicated instead of/before the abbreviation.  

C9) In the conclusion section, the authors mentioned about the general information, instead conclude their work. It would be better to be re-written with the conclusion of their results.   

Author Response

Dear Editor-in-Chief: Materials,

It gives me pleasure to resubmit our manuscript (materials-816813) titled ''Efficacy of bioactive glass nanofibers tested for oral mucosal regeneration in rabbits with induced diabetes'' which I am submitting for exclusive consideration of publication as an article in the materials, special issue of “Biomaterials for Dental Healing”.

We improved our manuscript based on the reviewers comments. All changes from the previous submission, marked with blue colour as per resubmission instructions

Please find attached file, the reply for reviewer’s comments.

Thank you for your time and consideration.

Answers to reviewers comments

Materials 816813

Reviewer 1:

Main comment

Answer

C1

Line 33: A space needed between and3. There are so many tied letters in the manuscript, especially before/after the brackets. It would be better to check it all.

Thanks. Modified

C2

Lines 51, 81, 114, 216 : The (IDF), (ROS), (Larry Hench, 1969), (3R) does not require brackets since all were defined previously.  

Thanks. Modified

C3

Line 88 : The D here supposed to be written in lower case “Furthermore, Diabetic patient…

Modified

C4

In Figure 1d, PVA is written, however, polyvinyl butyral  was reported in section 2.1 as used material. It needs to be fixed for both figure and text, either the used material was polyvinyl butyral (PVB) or polyvinyl acetate (PVA).   

Thank you for your comment.

The material used was ( poly vinly butyral)(PVB)

It was fixed in figure 1d

C5

Lines 191 and 352 : The sample name of “ 70SBG” supposed to be the same sample with the sample of 70S 30 Ca BG. One of these needs to be fixed.

70S 30Ca  was fixed

C6

Section 3.1.1 : The obtained peak corresponding to the Si-O-Si network, indicates porous structure (line 321) or amorphous structure (line 331) ? This need to be fixed and a reference needed as an indicator of the presence of Si-O-Si is indicating either porosity or amorphous structure. Also, “t0” supposed to be “to” (Line 330).

The authors are thankful to the Reviewer for pointing out the mistake. Herein the word ‘porous’ has been replaced by ‘amorphous’ in the revised manuscript . Relevant reference has also been incorporated.

C7

In Figure 4b, the scale shows 1um and if it is correct, the R is bigger than 560 nm, compared to the scale.  This needs to be checked to correct either scale or the R value of the fiber, which looks slightly bigger than 2 um.

Thank you for your comment.

The range of nanofibers diameter was fixed in the text and reference was added. In figure (4b) due to fibres aggregation the single fibre border was missed. So, a single fibre was marked and measured).

C8

Line 628 : The SBF, supposed to stand for the simulated body fluid. If so, it needs to be indicated instead of/before the abbreviation.

Thank you. Fixed

C8

In the conclusion section, the authors mentioned about the general information, instead conclude their work. It would be better to be re-written with the conclusion of their results.   

Thank you for your comment. The conclusion section was completely modified

Reviewer 2:

Main comments

Answer

The abstract fails to include a summary of the results of the in vitro component of the study; although, if this is okay with the editor, it is also okay with me, as the principal findings of the animal model are given.  The introductory information is useful, and provides a sufficient background for the general reader (or new-comer to the field) to appreciate the context of the study. The methods section is generally easy to follow; although, details of the reagents and concentrations used to prepare the glass must be included.

Thanks for your valuable comment.

Invitro characterization results were mentioned in the abstract as recommended.

Reagent and their concentrations were added

C1

Where the authors cite studies, they must include the numerical reference: e.g. Line 58 should read, ‘Saini et al. [4]’; and ‘work by Dey’ in Line 124 requires a reference. I think the authors are referring to the work of Delbert Day here, rather than ‘Dey’?

Thank you for your comment.

The references and the author name were modified as recommended.

C2

The authors need to subscript the numbers of the chemical formulae in Lines 131 and 594.

Modified

C3

All abbreviations, such as ‘XRD, FESEM’ (Line 133) must be defined prior to use. All abbreviations that have been defined, such as ‘VEGF’ (Line 100), don’t need to be defined again in later sections.

 Thank you for your comment.

Modified

C4

It would be good if the authors would provide a more explicit description of the work at the end of the introduction. The reader is forced to read the materials and methods section in depth if the aims and objectives are not comprehensively summarised at the end of the introduction. This is time-consuming and annoying - it is better to give the informed reader the option to skip directly to the results section and then refer back to the methods to check the specific details of any techniques or specific operating parameters if required.

The aim of the study was summarized in the introduction section as following:(line----- to ------ modified version)

In the present study, we aimed to synthesize a novel bioactive glass composition of (1-2) mole% of B2O3, (68-69) mole% of SiO2, and (29-30) mole% of CaO via the low-temperature sol-gel route (acronym: BG, bioactive glass) and fabricate BG nanofibre (BGnf) via electrospinning technique. Following fibers preparation, the in vitro physicochemical characterization (X- ray diffraction (XRD), field-emission scanning electron microscope (FESEM)) and degradation study were carried out to evaluate the scaffold bioactivity. . Moreover, assessment of the efficacy of BGnf in enhancing oral mucosal wound healing and angiogenesis was tested via histological and immunohistochemical analysis in rabbits with chemically induced type I diabetes mellitus.

C5

Can the authors please inform the reader, in the introduction, why they have elected to use this particular composition of bioactive glass? In particular, they should state the reason why they have chosen to add borate ions to the calcium silicate system. They don’t inform the reader of the significance of this until Line 623. This aspect of the study should not remain a mystery until the discussion.

This comments has been taken into considerations

It was included in the introduction section as following:

Between different types of bioactive glass, silicate and borate-based BG has significant importance in different soft and hard tissue applications. Despite that, borate BG is shown to exceed the silicate BG in the level of bioactivity. As it reacts with surrounding body fluids four to five times faster than silicate-based BG which allows for complete conversion of the glass into hydroxyl carbonate apatite layer (HCA) and intimate contact between the scaffold and the surrounding tissues. Thus, borate containing BG is more favourable to use particularly in soft tissue applications.

C6

The authors must provide a key for the chemical abbreviations used in Figure 1, and they must also provide the quantities of reagents used.

The key and quantities of the glass composition used are as following:

Briefly, glass sol was prepared by adding 0.680g Polyethylene oxide (PEO), 6 ml of tetra ethyl ortho silicate (TEOS), 3 ml of tributyl borate( TBB) and 6 g of calcium nitrate tetrahydrate (CaNO3.4 H2O) sequentially  into1N  HCl solution.

C7

SEM operating parameters should be included at the end of Section 2.2 (along with any sample preparation or coating/sputtering prior to imaging).

SEM operating parameters and sample preparation was as following:(fixed in the text)

Prior to imaging, glass fibers were sputter coated with a12 nm thickness of gold. An accelerating voltage of 10-20 kV and working distances of 5-8 mm were used.

C8

The origin of the calcium silicate bioactive glass (BG) listed in Table 1 must be mentioned in Section 2.3.

Thank you for your variable comment. The material was removed from the context of the draft

C9

Standard deviations or confidence intervals must accompany the concentration data listed in Table 1. Why is the calcium silicate glass included in this table? The authors should explain the relevance of this.

Standard deviation has been included in the revised manuscript.

The material was removed from the context of the draft.

C10

 The discussion seems to contain quite a lot of general information rather than focusing on a discussion of the specific results obtained in this study. Can the authors please try to tailor the discussion to an evaluation of their results in the context of the current literature, rather than broadly reviewing the subject area?

Thank you for your comment.

The discussion was tailored to contain the relevant information only without affecting the context

C11

79 references seem excessive for a small study such as this. Additionally, many of the references are over a decade old. Are they all necessary, or can the authors hone them down to just include essential contemporary papers?

Some reference contains the basic information regarding the mucosal graft and diabetes mellitus. The unneeded references have been removed.

Reviewer 2 Report

This is an interesting study on the application of an electrospun bioactive glass scaffold to promote oral soft-tissue healing in rabbits with Type I diabetes. The manuscript is enjoyable to read (although, the English language, grammar and formatting require extensive revision by a native speaker prior to publication). The study only includes one formulation of bioactive glass, and so the results are slender, yet they are sufficiently novel to warrant publication.

The abstract fails to include a summary of the results of the in vitro component of the study; although, if this is okay with the editor, it is also okay with me, as the principal findings of the animal model are given.  The introductory information is useful, and provides a sufficient background for the general reader (or new-comer to the field) to appreciate the context of the study. The methods section is generally easy to follow; although, details of the reagents and concentrations used to prepare the glass must be included.

  1. Where the authors cite studies, they must include the numerical reference: e.g. Line 58 should read, ‘Saini et al. [4]’; and ‘work by Dey’ in Line 124 requires a reference. I think the authors are referring to the work of Delbert Day here, rather than ‘Dey’?

  1. The authors need to subscript the numbers of the chemical formulae in Lines 131 and 594.

  1. All abbreviations, such as ‘XRD, FESEM’ (Line 133) must be defined prior to use. All abbreviations that have been defined, such as ‘VEGF’ (Line 100), don’t need to be defined again in later sections.

  1. It would be good if the authors would provide a more explicit description of the work at the end of the introduction. The reader is forced to read the materials and methods section in depth if the aims and objectives are not comprehensively summarised at the end of the introduction. This is time-consuming and annoying - it is better to give the informed reader the option to skip directly to the results section and then refer back to the methods to check the specific details of any techniques or specific operating parameters if required.

  1. Can the authors please inform the reader, in the introduction, why they have elected to use this particular composition of bioactive glass? In particular, they should state the reason why they have chosen to add borate ions to the calcium silicate system. They don’t inform the reader of the significance of this until Line 623. This aspect of the study should not remain a mystery until the discussion.

  1. The authors must provide a key for the chemical abbreviations used in Figure 1, and they must also provide the quantities of reagents used.

  1. SEM operating parameters should be included at the end of Section 2.2 (along with any sample preparation or coating/sputtering prior to imaging).

  1. The origin of the calcium silicate bioactive glass (BG) listed in Table 1 must be mentioned in Section 2.3.

  1. Standard deviations or confidence intervals must accompany the concentration data listed in Table 1. Why is the calcium silicate glass included in this table? The authors should explain the relevance of this.

  1. The discussion seems to contain quite a lot of general information rather than focussing on a discussion of the specific results obtained in this study. Can the authors please try to tailor the discussion to an evaluation of their results in the context of the current literature, rather than broadly reviewing the subject area?

  1. 79 references seem excessive for a small study such as this. Additionally, many of the references are over a decade old. Are they all necessary, or can the authors hone them down to just include essential contemporary papers?

Author Response

(The authors gave the same response as above.)
